# Embodied Semantics: Early Simultaneous Motor Grounding in First and Second Languages

**DOI:** 10.3390/brainsci14111056

**Published:** 2024-10-25

**Authors:** Juliane Britz, Emmanuel Collaud, Lea B. Jost, Sayaka Sato, Angélique Bugnon, Michael Mouthon, Jean-Marie Annoni

**Affiliations:** 1Department of Psychology, University of Fribourg, 1700 Fribourg, Switzerland; juliane.britz@unifr.ch (J.B.); emmanuel.collaud@cabinet-hi-mind.ch (E.C.); sayaka.sato@unifr.ch (S.S.); angelique.bugnon@etu.unige.ch (A.B.); 2Department of Neuroscience and Movement Science, Faculty of Science and Medicine, University of Fribourg, 1700 Fribourg, Switzerland; lea.jost@h-fr.ch (L.B.J.); michael.mouthon@unifr.ch (M.M.)

**Keywords:** language, embodied semantics, brain, ERP, bilingualism

## Abstract

Background/Objectives: Although the embodiment of action-related language is well-established in the mother tongue (L1), less is known about the embodiment of a second language (L2) acquired later in life through formal instruction. We used the high temporal resolution of ERPs and topographic ERP analyses to compare embodiment in L1 and L2 and to investigate whether L1 and L2 are embodied with different strengths at different stages of linguistic processing. Methods: Subjects were presented with action-related and non-action-related verbs in a silent reading task. Subjects were late French–German and German–French bilinguals, respectively, and we could therefore collapse across languages to avoid common confounding between language (French and German) and order of acquisition (L1, L2). Results: We could show distinct effects of embodiment and language. Embodiment affected only the sensory and lexical stages of processing with increased strength and power of the N1 component for motor vs. non-motor verbs, and language affected the lexical and semantic stages of processing with stronger P2/N400 components for L2 than for L1. Non-motor verbs elicited a stronger P1 component in L2. Conclusions: Our results suggest that processing words in L2 requires more effortful processing. Importantly, L1 and L2 are not embodied differently, and embodiment affects early and similar stages of processing in L1 and L2, possibly integrating other process of action–language interaction

## 1. Introduction

Semantic processing encompasses our conceptual knowledge of the world—the meaning of words, objects, and factual knowledge. It is central to human cognition and lies at the interface between language, memory, and perception. Former traditional models of semantic memory have considered semantic concepts to be represented amodally in a “mental lexicon” [1], which consists of multiple layers of representation, much like a real lexicon. In subsequent word production, for example, the word form representation represents the physical, i.e., phonological and orthographic features of a word, and the lemma level represents the grammatical feature level. The biggest caveat of the mental lexicon is the problem of recursion, namely, how to refer to a word by referring to other words (the symbol grounding problem) [2]. A solution to this problem is provided by theories of embodied cognition that consider semantic concepts not as abstract notions but as a result of the interaction between the body and the environment.

The term “embodiment” refers to the grounding of cognition in systems involved in low-level perceptual and action information processing. According to the experimental evidence of Pulvermüller, Hauk, Tettamanti, and others, embodied theories of cognition claim that higher cognitive processing, including language, activates the same brain sensorimotor structures involved when experiencing the environment [3,4,5,6]. One of the precursors of embodiment theories comes from neurophysiological investigations of noun and verb processing, which provide distinct cortical topographies as biological counterparts of words (e.g., matched nouns and verbs) as early as 200 ms after word presentations [7]. Subsequently, Hauk et al., using event-related fMRI, showed that action words referring to face, arm, or leg actions (such as lick, pick, or kick), presented in a passive reading task, differentially activated areas along the motor strip that either were directly adjacent to or overlapped with areas activated by actual movement of the tongue, fingers, or feet [5]. Such biologically grounded cognitive theories have led to the development of brain-constrained neural models [8]. In this approach, the sensory–motor system has been shown to be involved in the execution and comprehension of actions, and there is ample evidence that the sensory–motor system is crucially involved in language comprehension and production [4,9]: action verbs recruit somatosensory and motor cortex in a roughly somatotopic fashion through Hebbian learning [3,10,11,12]. Most fMRI studies on action language actually found activation in the motor and premotor cortices in accordance with the effector to which the stimuli refer to. One of the first functional Magnetic Resonance studies in this respect conducted by Hauk et al. showed how action verbs referring to the face, arm, or leg (e.g., to lick, pick, or kick) presented during a passive reading task activated the same areas appointed to the execution of the actions with the face, arm, and leg respectively. Tettamanti et al. found a similar result with sentences involving actions during a listening task. Boulenger et al. confirmed such mechanisms by comparing idioms and literal sentences, including arm- and leg-related action words [3,5,13].

There are thus a number of findings showing that the motor system is involved in language comprehension and that language is at least partially embodied. This is supported by vast anatomical and functional connections linking the motor system with language-related regions in the left inferior frontal (“Broca’s area”) and superior temporal (“Wernicke’s area”) [14,15] areas. It remains a matter of debate whether embodiment affects earlier (sensory and lexical) and/or later (semantic) stages of language processing, particularly concerning word recognition, which is the most used paradigm in the embodiment literature.

The electro-cortical time course of word recognition has shown several components linked with stages of word recognition, which have been classified as early and late components. As it will be outlined here, P1, N1 and P2 (or P200) are considered as early, “pre-semantic”, components, while N400 is considered a late component, a hallmark of semantic processing.
Early ERP components (from 50 ms up to ~150–200 ms after stimulus onset) are considered to reflect sensory and perceptual processes and vary with physical stimulus properties. In the case of word recognition and reading, the early components reflect the visual analysis (P1, generally occurring 100ms after stimulus presentation). The amplitude of the P1 component is particularly modulated by physical characteristics of words, such as font size, contrast, and their interaction, but also sometimes to emotional content [16]. Then, the contextual automatic language process, such as the early stages of visual word recognition and lexical processing, is reflected in the negative going potential component (N1, peaking at around 170 ms), [17]. This Time Window is particularly sensitive to letter and word recognition [16]. In particular, N1 peaked earlier to letters tather than pseudo letters according to [16], and such a component is also sensitive to attentional charge. In addition, while the N1 component is generally linked to early sensory and prelexical processing, it has also been associated with early lexical activation [18,19]. Different studies indicate that the N1 comprises multiple subcomponents [20], with its later phase showing sensitivity to lexical processing [17]. According to Eberhard-Moscicka, adult speakers of English as a foreign language exhibited print tuning in the early N1 and lexicality effects in the late N1, indicating that sensitivity to print and lexicality unfolds differently throughout the N1 segment in adults [17].

An intermediate electric activity, starting with a P200, a positive going potential component, has been related to orthographic and phonological pre-lexical processing well as to lexical access [15].
2.The later ERPs (after ~200–250 ms post-stimulus) are insensitive to physical variations of the stimulus and vary with respect to cognitive processes [21,22]. Notably, the N400, with a centro-parietal scalp distribution, reflects semantic judgment and integration during processing of the presented words, where amplitude increases negatively following semantic anomalies [23,24,25,26]. Originally, this event-related potential (ERP) was seen as a signature of semantic incongruity, but, since then, other types of semantic tasks (e.g., semantic contextual integration, processing of single words out of context, tasks involving a concreteness effect) have shown the same robust signature [22,27]. For these reasons, N400 is considered a hallmark of semantic processing. Bilingual subjects do recognize L2 words more slowly compared to L1, and the neurophysiological correlates of this target language difference are generally detected in the early rather than the late stage [28,29].

Physiological measures with high temporal resolution, such as electro-(EEG) and magnetoencephalography, and, to a lesser degree, transcranial magnetic stimulation applied at different time points, have yielded timing information about the involvement of motor areas in language processing. Some studies using single pulse Transcranial Magnetic Stimulation, a technique which can influence a behavioral process in the range of tens of milliseconds [30]. Tomasino investigated semantic resonance (motor-to-language directional effects) with this technique and found embodiment effects within 200 ms, suggesting that it may take place at early stages of word recognition [31]. It is also possible to process semantics, e.g., by dissociating word categories (action vs. object) at about 150 ms and in modality-specific areas [32]. Hauk and Pulvermuller investigated early electrophysiological differences between matched semantic categories of action words in a passive reading task; significant differences between subcategories of action words were present at approximately 220 ms [33]. On the other hand, Buccino et al. showed that listening to action-related sentences with hand or foot effectors modulates the activity of the motor system about 500–700 ms after the beginning of the sentence, suggesting that motor resonance of words may occur later, during semantic processing [34]. In summary, while some data suggest that embodiment affects later semantic stages of processing (see also [35]), the majority of evidence points toward early effects of embodiment within 200 ms after stimulus onset [13,32,33,36]. It is, however, difficult to disentangle the possible roles of the motor cortex and other lexical characteristics, such as word frequency and physical orthographical and lexical integration correlates, taking place within 200 ms [27].

Most studies have investigated embodiment only in the mother tongue (first language/L1), and less is known about whether and at what stage a language learned later in life (L2) might recruit motor areas [37]. Generally, the term Age of Acquisition (AoA) refers to the age at which one begins to learn an additional language, which may or may not coincide with the age of arrival in the context of that language [38]. AoA has been used to classify bilinguals into simultaneous (AoA = from birth), early/childhood (AoA = prior to age 7 or 12, depending on the studies), and late/adult bilinguals (AoA = after the age of 7 or 12, or post-puberty) (e.g., [38]). In our research, we classify as late bilinguals the participants who learned L2 after 7 years of age [39]. The L1 is acquired implicitly and through interaction with the environment, and the acquisition of semantic concepts is assumed to be a direct consequence of that interaction with the environment. According to experimental data and theoretical models, representations of concepts in the first (L1) and second language (L2) partially overlap, particularly in early bilinguals [39,40]. However, in late bilinguals, L2 vocabulary is often acquired in late childhood and typically in a classroom context (particularly when learnt in secondary school), which is mostly based on explicit memory. In Switzerland, German and French are the most important national languages and are formally learned, respectively, by L1 French- and German-speaking children at school in the 5th grade. There is no immersive learning at this stage. Therefore, the question arises as to whether sensorimotor experience is involved in late L2 learning concepts. Based on this assumption, we might suggest semantic representations in L2 to be less embodied than in L1 [39]. One could argue that the L1 acquired early in life through interaction with the environment is embodied more strongly than an L2 acquired later and through formal instruction and not in an immersion context. Given that L1 and L2 are acquired differently, one could assume that the embodiment of L1 affects both early/lexical and later/semantic stages but that embodiment affects only later/semantic stages in L2 because the lexical representations in L2 are learned formally and without interaction with the environment and then mapped onto stored semantic concepts that were themselves acquired through interaction with the environment during L1 acquisition.

Several behavioral and neurophysiological studies provide evidence for the notion that L2 is indeed less or differently embodied than L1. Prepositions denoting positions in space appear to be more strongly embodied in L1 than L2 [41], power words are likewise more strongly embodied in L1 than L2 [42], negative words are more prone to disembodiment in L2 than L1 [43], and there is increased semantic resonance in L1 than L2 at early latencies and increased motor resonance in L2 than L1 at mid-latencies [44,45]. In an interesting recent study, Zhang et al. recorded mu activity induced by the presentation of written action words in Chinese L1 and in English L2 action verbs. The results showed that, compared to abstract words, action verbs induced higher early motor activation in both languages, as indicated by the desynchronization of the EEG mu rhythm at 250 ms after stimulus presentation [46]. Their data also suggest that the mu-event-related desynchronization was stronger in L1 than in L2.

Functional neuroimaging studies, on the other hand, provide a different image, namely that L1 and L2 are embodied similarly. The first fMRI study comparing embodiment in L1 (German) and L2 (Dutch) found comparable recruitment of motor areas in L1 and L2, irrespective of whether the words were cognates or not [47], suggesting that semantic representations in L2 are sufficiently rich to activate motor regions. Similarly, action verbs presented in L1 and L2 recruit the primary motor cortex and the precentral gyrus in L1 (Mandarin) and L2 (English), although, in L2, the semantic integration hub was connected less strongly to sensorimotor regions [48]. In a similar vein, motor areas are activated more strongly for motor versus non-motor verbs and more strongly for L2 versus L1 verbs, but there was no difference for motor verbs in L1 versus L2 [49,50]. These recent fMRI studies provide converging evidence that embodiment is comparable in L1 and L2, but due to the slow time course of the BOLD response, they cannot answer the question of whether embodiment affects earlier or later stages of linguistic processing or whether L1 and L2 might be embodied at different stages of processing.

Distinguishing between early and later stages of processing requires a continuous physiological measure with high temporal resolution, such as the electroencephalogram (EEG). The EEG represents the temporal evolution of the electrical field generated by the brain and is represented by the topographic configuration of the scalp field. Event-related potentials (ERPs) are extracted from the ongoing EEG by making a time-locked average across multiple stimuli of an experimental condition, thereby retaining the brain activity evoked by stimulus processing and averaging out any EEG activity not directly related to stimulus processing.

In the present study, we employed the high temporal resolution of ERPs and global ERP measures to compare the time course of embodiment of action verbs in L1 and L2. We separately manipulated language (L1, L2) and motor relatedness (action verbs, non-action verbs) and recorded the EEG while subjects performed a silent reading task with an embedded semantic judgment task of written words. We focused on words that were known by the participants. This design allows us to investigate the time course of signatures of embodiment separately for L1 and L2 and whether they interact at some point. This can answer the questions of whether embodiment affects earlier or later stages of processing, whether this is observed both for L1 and L2 and whether there are separate or interactive effects of embodiment and language, i.e., whether action words are embodied differently in L1 and L2 in terms of strength and/or time course.

## 2. Materials and Methods

### 2.1. Stimuli

#### Pre-Tests

The stimuli used for the experimental paradigm were 200 verbs composed of action verbs, referring to motor actions executed by the hands (for example, “grasp”) and non-motor verbs (for example, “guess”), in French and German. Two pre-tests were conducted to ensure that the experimental stimuli fulfilled the desired criteria, namely that the hand–motor-related verbs were indeed judged as motor-related and that the hands were exclusively recruiting for their execution and that the non-motor-related verbs were indeed not motor-related and that they were likely to be known to most subjects in their respective L2.

The first pre-test evaluated the motor-relatedness, embodiment, valence, and emotional experience related to the stimulus material. A total of 361 French and German verbs were randomly split into two lists that were administered to naïve university students with a mean age of 21.55 years (SD: 2.44) who were native speakers of French (*n* = 31, 2 males) and German (*n* = 41, 5 males) so that each participant had to evaluate only one list of 181 verbs. Subjects had to rate (1) the level of embodiment, (2) the emotional experience, and (3) the valence of that verb on a 7-point Likert scale. They had to furthermore indicate whether this verb represented a movement of the body and, if so, which part of the body (head, face/mouth, one hand/arm, both hands/arms, one foot/leg, both feet/legs). Verbs considered to be too ambiguous to be classified as motor-related were excluded. Embodiment ratings were generally higher for French than German words (French motor verbs: mean = 5.02, SD = 0.74; French non-motor verbs: mean = 1.52, SD = 0.47; German motor verbs: mean = 4.53, SD = 0.76; German non-motor verbs: mean = 1.73, SD = 0.43), and thus the values were transformed into z-scores, but this difference was not significant.

The second survey was performed to select verbs that would likely be familiar to the participants in their second language. This step was necessary as we finally included in the ERP analysis only the neurophysiological correlates of verbs that were known to the participants to make sure that the words were semantically processed. The same list of 361 French and German verbs that was used in the first pre-test were administered to another group 26 German native speakers (age M = 22 years, SD = 2 years) and 36 French native speakers (age M = 22 years, SD = 3 years) who had learned German and French as an L2 after the age of 7. Subjects were asked to rate their familiarity with each word on a 7-point Likert scale, and items with both a median and a mode below 4 were excluded; this led to the exclusion of 103 French and 136 German verbs.

### 2.2. Final Stimulus Set

The final stimulus set comprised 200 verbs, with 50 in each condition (French motor, French non-motor, German motor, German non-motor), and they did not contain any cognates (The list can be loaded in the Appendix A). All verbs were matched for lexical frequency, number of letters and syllables, and valence, which did not differ between languages, motor-relatedness, or both factors. Motor-relatedness was rated significantly higher for the motor versus the non-motor verbs (F(1,1) = 1141.21, *p* < 0.001) but not for the languages (F(1,1) = 1.97, *p* > 0.05), and it was not rated differently between the languages (F(1,1) = 1.83, *p* > 0.05). Likewise, the emotional content was rated higher for non-motor versus motor verbs (F(1,1) = 50.48, *p* < 0.001) but not for the languages (F(1,1) = 1.77, *p* > 0.05), and it was not rated differently between the languages (F < 1). In addition to the 200 experimental verbs, a total of 100 filler nouns (50 German, 50 French, half animate, half inanimate) were added to the experimental lists to render the experimental conditions opaque.

### 2.3. Subjects

Our a priori power analysis indicated that a 2 × 2 repeated measure ANOVA with the parameters α = 0.05, f = 0.25, 1-β = 0.8, r = 0.5 between measures, and ε = 1 requires a total sample size of n = 24 (G*Power) [51]. As such, 29 subjects (10 male) with a mean age of 23.2 years (SD 4.27, range 19–40) participated in the EEG experiment for monetary compensation (CHF 20/h).

None of them had any history of neurological or psychiatric disease and all had normal or corrected-to-normal visual acuity and were right-handed. They had either a high school diploma (Matura) or a university degree (Bachelor’s or Master’s), which is a minimum of 13 years of education. Fribourg is a bilingual university, and participants attended lectures and seminar in French or German, but some courses were also in English. Our experimental group included native German-speaking and native French-speaking students. Inclusion criteria for the study were that L1 German-speaking participants had completed primary school in the German part of Switzerland and that they were exposed only to Swiss German and/or German before the age of 7 and that the L1 French group was only exposed to standard French before the age of 7. According to Swiss schooling rules, the L2 language was French for the German group and standard German for the French group. Immersive second language learning program in the other language before the end of primary school as well as early bilingualism were exclusion criteria. The simultaneous or later third language was generally English in both groups. Fifteen were L1 speakers of French and learned L2 German after the age of 7 at the average age of 8.4 (SD: 1.12) years, and fourteen were L1 speakers of German and learned L2 French at the average age of 10.78 (SD: 2.35) years. To control for language, participants with French and German as their mother tongue were pooled into one group such that German and French were equally represented in L1 and L2 as mother tongues. All were right-handed according to the Edinburgh Handedness Inventory [52], with an average score of 91.14 (SD: 12.60). The study was approved by the Ethical Committee of the Canton of Vaud, CH, and written informed consent was obtained prior to participation.

### 2.4. Language Evaluation

Before the EEG experiment, the objective L2 receptive vocabulary was assessed with the vocabulary sub-test from the computer-based DIALANG language diagnosis system [53]. This lexical decision task gives a score from 0 to 1000 and allows for categorizing the L2 knowledge of the participants using the following scoring system: 0–100, knowledge of very few words; 101–200, very basic knowledge; 201–400, a limited vocabulary; 401–600, a good basic vocabulary; 601–900, an advanced level with a very substantial vocabulary; and 901–1000, a native speaker level.

After the silent reading task, subjects performed a translation recognition task of the same items. This was performed to include in the analyses only those verbs that were known to the subjects. Every experimental verb item was presented with four translation options, plus the option “I don’t know”. Subjects were asked to indicate the correct translation, and in case they did not know the word, they were asked to refrain from guessing and instead directed to choose “I don’t know”.

We analyzed participants’ accuracy in the semantic judgment task based on both the filler items and the translation recognition task.

### 2.5. Procedure

#### Experimental Task

The 200 experimental stimuli and 100 filler words were divided into two lists. Each list consisted of 40 short blocks of 8 stimuli each. Within each block, all stimuli were of the same language, with equal numbers of motor- and non-motor-related verbs, which were randomized. This was done to foster a monolingual context and to avoid language switching within a block.

Figure 1 depicts the experimental procedure. All stimuli were presented in white on a black background for 800 ms, with an interstimulus interval varying randomly between 2000 and 3500 ms, and the subjects were instructed to silently read the words presented on the screen. In order to ascertain that the subject actually effectuated the task and semantically processed the stimuli, they were asked to occasionally perform a semantic judgment on two filler items (verb + nouns, e.g., “swim”, “fish”). E-Prime 3.0 was used both for stimulus presentation and response collection in the semantic judgment task. Between blocks, subjects could take self-paced breaks, and the task lasted roughly 25 min. Filler items were not included in the ERP analyses.

### 2.6. EEG Acquisition and Data Reduction

The EEG was acquired from 128 active Ag/AgCl electrodes (Biosemi Active-Two, Biosemi B.V., Amsterdam, The Netherlands) with a sampling rate of 1024 Hz. Before segmenting the EEG into epochs, the ongoing EEG was bandpass-filtered between 0.5 and 40 Hz using a 2nd order Butterworth filter with a −12 db/oct roll-off. The filter was computed linearly with two passes (one forward and one backward), thus eliminating the phase shift, and with poles calculated each time to the desired cut-off frequency. ERP epochs of 800 ms encompassing a 100 ms pre-stimulus baseline were extracted for each participant and for each experimental condition. Epochs were visually inspected for ocular–motor, myogenic, and other artifacts, and trials contaminated with noise were rejected.

### 2.7. Topographic Analysis of Event-Related Potentials

Local EEG and ERP measures consider the time course of amplitude modulations and global EEG measures considering the time course of topographic differences between conditions. Several aspects make it difficult to interpret the functional significance of local amplitude differences. Because local amplitudes are reference-dependent, their location can vary with the reference, and different references can make effects appear or disappear [54]. Moreover, amplitude differences can arise from (I) strength differences of the same neuronal generator, (II) latency differences of the same neuronal generator, or (III) different generator configurations. Local amplitude measures cannot distinguish between those cases, which renders their functional interpretation arbitrary at best.

This is why we adopted global, topographic analyses of EEG/ERPs. They are a solution to this problem because they consider the (global) topography of the scalp electrical field, which is independent of the reference. Different references affect the voltage potential between a given local electrode and the reference, but this fact does not affect the brain activity generating the scalp field. Because topographic differences necessarily indicate different intracranial generators [55], global EEG/ERP measures can distinguish between differences in (I) strength, (II) latency, and (III) configuration of the underlying sources and hence elucidate the timing with which either the same or different processes unfold at a given time. Topographic ERP measures consider ERP components not as time-series of local peaks and troughs but as time-series of stable map configurations that represent the different stages of processing [56,57,58].

To investigate whether embodiment affects (I) early or later stages of processing, (II) whether this differs in L1 and L2, and (III) whether there are separate or interactive effects of embodiment and language, we compared the stimulus-evoked potentials for the motor- and non-motor-related verbs in L1 and L2 using a global measure. To that end, we submitted the grand-average ERPs in the four experimental conditions to a spatio-temporal segmentation procedure implemented in the Cartool Software version 3.70 by Denis Brunet [56,57,58] to identify template maps representing the sequence of dominant topographies in the different conditions. This procedure identified whether the same or different scalp topographies are evoked in the experimental conditions [58,59,60,61] and follows the notion that stimulus-evoked topographies do not vary randomly from time point to time point but that they remain stable for discrete processing stages with sharp transitions between states representing the stages.

According to our hypotheses, the periods of interest should correspond to P1, N1, and P2 components during the early processing phase of word recognition and the N400 component for the later “semantic” phase. To determine them without bias, a data-driven approach was applied. In order to identify the dominant topographies, we used a spatial atomize–agglomerate hierarchical cluster analysis (AAHC) [58,59,60] where the topography was normalized with respect to the field strength (Global Field Power (GFP)), which is the spatial standard deviation of the scalp electrical field. This procedure identifies dominant topographies based solely on their configuration and irrespective of their strength. Importantly, we did not restrict our cluster analysis to identifying a given number of templates or explaining a certain amount of variance. We computed 20 different solutions of the AAHC analysis and identified the optimal solution through the convergence of a combination of multiple criteria [62] that attempt to maximize the explained variance with a minimum number of clusters. Because ERP map topographies of <10 ms are physiologically implausible, we applied a temporal constraint criterion of 10 ms.

After identifying the optimal solution of the cluster analysis, we determined how well each template identified in the cluster analysis on the grand-average was represented in each condition in each subject. We computed a strength-independent spatial correlation between each template map and the ERPs of every individual in all conditions using a winner-take-all assignment for the templates. Finally, we assessed whether the ERPs differed statistically between the experimental conditions concerning field strength (GFP), global explained variance (GEV, the explained variance weighted by the GFP), and microstate duration by means of a 2 × 2 repeated measures ANOVA with the factors *language* (L1 vs. L2) and *embodiment* (motor vs. non-motor verbs).

## 3. Results

### 3.1. Behavioral Results

L1 French speakers started learning German, on average, at 8.4 (SD = 1.12) years, and L1 German speakers learned to speak French, on average, at 10.78 (SD = 2.35) years; this difference was not significant (t(1,28) = −1.7141, *p* = 0.09).

Table 1 summarizes the DIALANG L2 proficiency scores and the performance in the translation recognition task for all subjects and separately for the L1 French and L1 German speakers. Even though the L1 German group had slightly higher overall scores than the L1 French group, none of these differences were significant (DIALANG scores: t(1,41) = 1, *p* = 0.322; recognition performance (all words): t(1,41) = 1.53, *p* = 0.13; recognition performance (motor verbs): t(1,41) = 1.1, *p* = 0.27; recognition performance (non-motor verbs): t(1,41) = 1.37, *p* = 0.17). Hence, we could collapse the L1 and L2 across French and German for the ERP analyses. According to the mean DIALANG performance, participants were considered to have a good basic vocabulary. The performance in the translation recognition task correlated highly with the DIALANG L2 scores (r = 0.79, *p* < 0.001) and equally for motor (r = 0.88, *p* < 0.001) and non-motor (r = 0.89, *p* < 0.001) verbs. Subjects correctly responded in 84.28% (SD: 11.36) of the semantic judgment trials, indicating that they indeed paid attention to the stimuli and processed them semantically.

### 3.2. Event-Related Potentials

The results and interactions are summarized in Figure 2 and Table 2. Illustration A of Figure 1 shows the different time windows of the maps identified through cluster analysis. Map 2, corresponding to the quasi totality of P1, runs from 100 to 150 ms. Maps 4, corresponding to N1, runs from 150 to 300 ms. Maps number 6, 7, and 8, corresponding to the N400 time window, extend essentially from 400 ms to 500 ms. The main effects of language were found both for the sensory (Map 2 (P1)), lexical (Map 5 (P2)), and semantic states of processing (Map 8 (N400)) both with respect to GEV and GFP, where all components were more strongly expressed in L2 than in L1. The main effects of motor relatedness were found for the sensory (Map 2 (P1)) and lexical stages of processing (Map 4 (N1)); the P1 was more strongly expressed for non-motor than motor verbs, and the N1 was expressed more strongly for the motor than non-motor verbs, but there were no effects of embodiment at the semantic stages of processing. Apart from P1, where Map2 had the highest GEV in the L2 non-motor condition and none of the other conditions differed, the factors never interacted, indicating that the effects of language and embodiment are separable and independent.

Moreover, the proficiency measures did not correlate with the ERP measures, and the number of words known to the subjects and the DIALANG scores were not correlated with the ERP map strength (GFP) or the explained variance (GEV).

## 4. Discussion

In the present study, we used topographic ERP analyses to delineate the time course of embodiment in the mother tongue (L1) and a language learned later in life (L2) in a formal/scholastic context. For both early sensory, mid-latency lexical, and late semantic stages of processing, we could identify separate effects of motor relatedness and of language. Apart from the earliest sensory stage of processing, these factors never interacted, signifying that in our paradigm, L1 and L2 known motor-related words do not show signatures of different embodiment. Our main research aims were (i) to replicate electrophysiological correlates of word embodiment in a silent reading task and (ii) to compare the time course of embodiment of action verbs in L1 and L2 and eventually detect differences in time and brain activity. The main results were the following: (i) grounding of action words occurs in early phase of word recognition at the level of the P1 map (corresponding to 100 ms after word presentation) and the N1 map (corresponding to 150 to 300 ms after word presentation, and (ii) this embodiment process is similar in time for both languages and associated with neural larger activity in L2 for non-motor verbs at the level of the P1 time window.

At the earliest stage of processing, we found a stronger P1 (100 ms after stimulus presentation) component in L2 compared to L1 and for non-motor versus motor verbs. This indicates that the early visual analysis of the verbal input is more effortful in L2 than in L1 and for the non-motor versus the motor verbs. It was more strongly expressed for non-motor verbs in L2 than all other conditions, indicating that sensory processing is most effortful in L2 abstract (non-motor) verbs. The N1 component (in our study, ~150–300 ms) was modulated by embodiment and more strongly pronounced in the motor than the non-motor condition. This corroborates the results from other studies that found stronger motor cortex activity for motor versus non-motor verbs in this time window in L1 [13,32,33,36,63]. Even if the embodiment process has a semantic resonance, most of the experimental pieces of evidence, including ours, suggest that it takes place early in the word recognition stage, possibly at the time of lexical processing. Some monolingual studies using a more implicit semantic paradigm have also suggested embodiment grounding later in the process [64], so we cannot exclude the modulation of such a grounding process in relation to the task processes. We actually could show here that the early embodiment process occurs in the first and second languages. Importantly, the N1 component was not modulated by language, it did not differ between L1 and L2, and there was no specific effect of embodiment between L1 and L2, which indicates that words are not embodied with different strengths in L1 and L2.

Both the P2 and the N400 were exclusively modulated by language, with both components more strongly expressed in L2 than in L1, indicating that semantic integration in our task was more effortful in L2 than in L1. These components were not modulated by embodiment and not differentially so in L1 and L2, which corroborates the finding that embodiment was restricted to lexical stages of processing, as indexed by the stronger N1 for motor and non-motor verbs, and that it did not intervene during semantic or conceptual stages of processing. With the high temporal resolution of ERPs, we show here that embodiment within this task exclusively affects sensory lexical stages of processing and equally so for L1 and L2, and that the more effortful processing of L2 selectively affects word recognition and the semantic stage of processing. This early impact of embodiment in the process of word recognition (P1 and N1) confirmed other ERP studies. Vukovic and Shtyrov, for example, examined mu–rhythm desynchronization as an index of motor cortex activity in response to L1 and L2 abstract and action prime–probe verb pairs. They found that cortical motor activation was present in both L1 and L2 around 150 ms post-stimulus [65]. Studies on novel word learning also showed early motor grounding effects, suggesting that such effects also occur early when learning a second language [66]. This furthermore indicates that once the word is stored in the semantic memory, embodiment does not differ between a language learned implicitly and through interaction with the environment early in life and another language learned later in life through formal instruction. This is in line with other studies that show how learning new words, languages, or concepts later in life can still lead to sensorimotor grounding [66].

Our results are in contrast with those from behavioral and neurophysiological studies that found a stronger embodiment of L1 than L2. Behavioral measures (reaction times, accuracy rates) only reflect the end-product of a cognitive process with a certain delay, and they cannot reveal which stage of processing contributes to the effect. In the case of embodiment, e.g., reaction times, measures cannot distinguish whether embodiment affects early or later stages of processing or whether embodiment affects different stages of processing in L1 and in L2. Recent data suggest that motor–language coupling effects seem to be present in both languages, although they can arise differently in each [67]. Concerning neurophysiological data, Xue et al. presented a case where L1-Ch L2-English participants with high (e.g., crumb) and low (e.g., lace) body–object interaction (BOI) English words obtained a slightly later embodiment effect [68]. These words were embedded in high (e.g., you brush the small sticky crumb) and low (e.g., you wear a string of cotton lace) sensorimotor contexts. It is possible that the impact on later time windows is due to the type of task, integrating also syntactic and sentence-related information. Our findings are in line with evidence from fMRI that shows no differences in embodiment but stronger recruitment of language areas in L2 than in L1 [47,48,50], indicating that L1 and L2 are embodied to comparable degrees. Due to the slow time course of the BOLD response, these fMRI studies cannot distinguish whether embodiment affects lexical or semantic stages of processing, or whether L1 and L2 might be embodied at different stages of processing. The stronger P1 component in L2 for non-motor during the P1 time window is also in contrast with other studies. If the strength of embodiment was reflected in this component, it should be more strongly pronounced in the motor than in the non-motor condition [69], which is not the case. One potential explanation for the stronger P1 (early visual potential) non-motor component in L2 could be related to access to abstract words in L2, as mentioned, or it may reflect interlanguage differences in ortho-phonological processing [70], with possible decreased typicity between perception and meaning in L2. Another explanation for such a stronger GFP in L2 at 100 ms could be explained by larger co-activation of the embodiment effect in the early stages of word recognition in L2. Access to L2 representations would require mediation via L1, especially in case of low L2 proficiency. This entails a later sensorimotor involvement when L2 proficiency is low compared to when it is high or compared to L1. Such differences in the degree of L2 embodiment would also be in line with the Dijkstra’s BIA+ model [71]. Overall, our data, in conjunction with other studies, point toward early embodiment of a late acquired L2, as similar in intensity to L1 embodiment, possibly integrating other processes of action–language interaction.

Other reasons for some divergences between studies are the differences in paradigms and in the constraints of the other studies. These studies also used stimuli not directly related to the execution of motor actions, like action verbs, but different paradigms for actions, e.g., prepositions denoting positions in space congruent or incongruent with a hand movement [41], power words [42], or words of different valence not related to executable actions [43], and may thus coin other aspects of embodiment. Also, the choice of a task that requires an overt motor response in every trial makes it hard to unequivocally attribute motor cortex recruitment or embodiment in general to the processing of the linguistic item or the execution of the response to this linguistic item [72]. Such explicit judgment may implicate less automatized processes if subjects have to judge whether a given item refers to a motor action or not [44]. In order to account for this confound, we opted for a silent reading task that does not require an overt response in every trial and ERPs, an unobtrusive measure that can identify at what stages of processing language and embodiment exert their influence and whether so independently or jointly with high temporal resolution. To assure that subjects processed the stimulus material semantically, they were asked to perform a semantic judgment on a small proportion of trials. Monaco’s study on the other hand, focused on semantic and motor resonance during reading of known motor words, thus focusing on excitability and connectivity of the motor cortex itself, in L1 and in L2 language mode rather than the embodiment process itself. Finally, a very interesting comparison can be done with a similar experiment done with Chinese/ English bilingual participants, where an early difference in the embodiment process was found between L1 and L2 in desynchronization of the EEG mu rhythm at 250 ms after stimulus presentation [26]. The time window corresponds to our “grounding-related “time window. This shows that some differences may be found. In our experiment, only correct responses were analyzed, avoiding effects related to the language itself by mixing both maternal French and German in the L1 group. So, our data suggest that the similarity in embodiment strength between L1 and late L2 is present only if the proficiency of words is similar. These differences between studies bring also indication that there may be some differences in embodiment according to the age of acquisition, but which concern internal electrophysiological processes rather than strength and duration of activation.

Our experimental design could exclude two of the main confounds, which are important caveats in studies comparing embodiment in L1 and L2, as follows: (I) the order of language acquisition (L1 vs. L2) with language-specific effects, such that L1 was always one language (e.g., Chinese) and L2 another (e.g., English), and (II) language with proficiency effects, by including all L2 words in the analysis irrespective of whether they were known to the participants or not, thus comparing a language in which all words were known (L1) with one in which only a subset of words were known and could thus be processed (L2). We could overcome the first confound by collapsing across languages and considering L1 and L2 irrespective of the [46] language. Being able to recruit subjects from the only official German–French bilingual university, participants in our study were L1 speakers of both French and German who acquired the respective other language as L2 (L1 French speakers had L2 German and L1 German speakers hat L2 French), thus excluding the important first confound. We could overcome the second confound by carefully pre-testing our material to exclude any differences in motor involvement in L1 and L2 and to use only words that are likely to be known to most of our participants. Importantly, and contrarily to other studies, we included in the analysis only those words that were known by each individual and that could be semantically processed. We can thus rule out that the confound between language and proficiency (namely that subjects are less proficient in L2 than L1 and that the analyses included words that could not be processed) could account for differences in embodiment between L1 and L2.

There are of course some limitations in our study. The first one is the number of participants. Although the number corresponds to the one calculated with our power analysis, it may be in the lower range of what is expected for fine modifications. The second potential limitation refers to the mother tongue. German-speaking Swiss participants are mostly diglossic (like in the Arabic linguistic context), with Swiss–German being a dialect spoken by the German-speaking community since birth in Switzerland and standard German being the official academic language [73]. However, at the word level, there is a strong similarity between Swiss German and standard German verbs, and this situation does not seem to influence word recognition. Finally, we did not quantified the linguistic distance, including characteristics such as overlaps in vocabulary, grammar, or script, which could modulate the extent of L2 embodiment, To the best of our knowledge, this issue was to a certain extent examined in a study by Ahlberg et al. (2017), which assessed L1 and L2 embodiment effects by comparing native German speakers with non-native German speakers whose L1 either similarly or distinctly encodes spatial relations. Although embodiment effects were found for both non-native speaker groups, these effects were not affected by the linguistic distance between their L1 and L2 [41]. These results speak to the fact that the extent of L2 embodiment may not necessarily correlate with the degree of overlap between the languages compared. However, we would argue that more research is necessary to examine such an issue.

## 5. Conclusions

These results shed new light on the time course of embodiment in L1 and L2. With the high temporal resolution of ERPs, we can show that embodiment affects lexical rather than semantic stages of processing both for L1 and L2 and that L1 is not embodied differently than L2, except for P1 short time windows, where L2 embodiment seems to require more effortful neural effort. We can furthermore show that L2 requires more effortful processing than L1, but mostly at semantic stages of processing. Our results are in line with recent fMRI studies that likewise show no differences in embodiment between L1 and L2. The slow temporal dynamics of fMRI, however, cannot distinguish whether embodiment affects different stages of linguistic processing in L1 and L2. In principle, L1 and L2 could be embodied at different stages, but with the high temporal resolution of ERPs used here, we can rule out that explanation. Overall, the first conclusions arising from this emerging topic of research point toward the early embodiment of a late-acquired L2 being similar to that of L1 embodiment, but it may possibly integrate other aspects of action–language interaction that deserve to be further delineated in their nuances.

## Figures and Tables

**Figure 1 brainsci-14-01056-f001:**
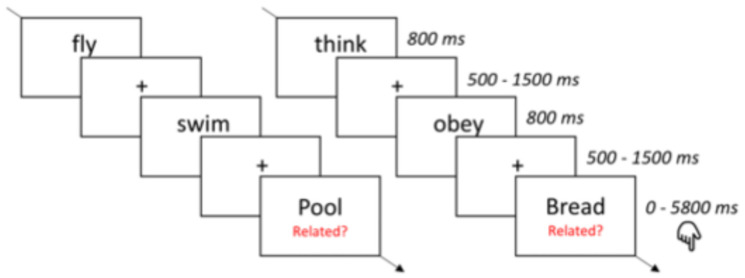
Experimental procedure. All stimuli were presented in white on a black background for 800 ms; the interstimulus interval varied randomly. Subjects read the words silently (see the text for a detailed description). Related pairs are presented on the left side of the figure, unrelated pairs on the right side.

**Figure 2 brainsci-14-01056-f002:**
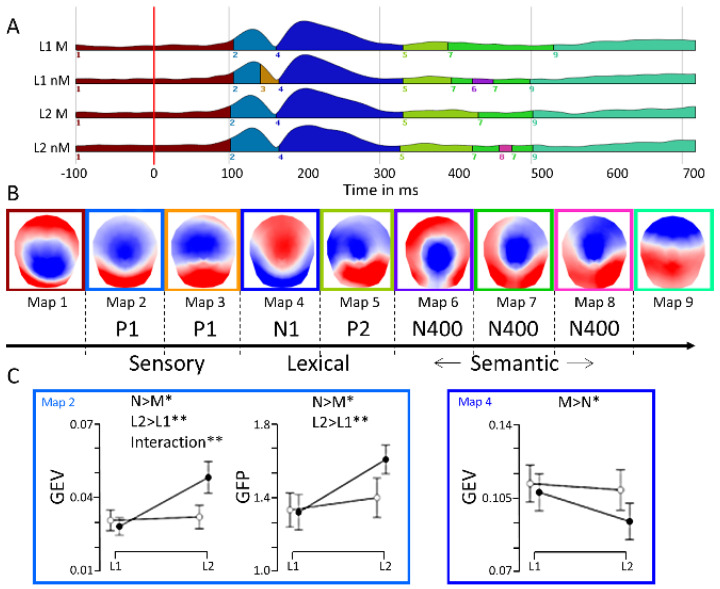
Results of the spatio-temporal segmentation procedure for the analyzed words. (**A**). Illustration of the computed segmentation for each experimental condition (amplitude of plots is the Global Field Power [GFP]). (**B**). Topographies of the microstates isolated. (**C**). Graphical illustration when the motor-relatedness factor is significant for the global explained variance (GEV) and GFP (details in Table 2). White dots = motor-related verbs (M), black dots = non-motor-related verbs (nM), * = 0.05 > *p* > 0.01, ** = *p* < 0.01. L1 = mother tongue, L2 = second language learned at a later stage. Map 2, which correspond to the largest part of the P100 time windows, extends between 100 and 150 ms after stimulus presentations. Map 4, which corresponds to N1, extends between around 150 and 300 ms after stimulus presentations.

**Table 1 brainsci-14-01056-t001:** L2 proficiency scores (mean/standard deviation) assessed using the DIALANG and the translation recognition task for the L2 verbs used in the EEG experiment. “L1 French” means the subgroup of participants with French as their maternal language and German as their second language. L1 German is the subgroup of participants with German as their maternal language and French as their second language. As mentioned in the text, both groups were merged to avoid language-specific bias.

	All Subjects	L1 French	L1 German
DIALANG L2 scores	445.3/252.6	377.3/246.8	512.1/228.4
Translation recognition in %			
All words	70.45/18.9	62.2/14.8	79.3/18.1
Motor verbs	70.48/19.9	64.0/17.7	77.4/20.5
Non-motor verbs	73.86/18.3	67.0/16.3	81.1/17.9

**Table 2 brainsci-14-01056-t002:** Statistical differences (ME = main effects for language, embodiment, and their interaction) in GEV and GEP for the maps, identified in the spatio-temporal segmentation procedure, where differences were found between conditions. Significant effects are highlighted in **italic**/**boldface**.

	Map 2 (P1)	Map 4 (N1)	Map 5 (P2)	Map 8 (N400)
GEV				
ME language	** *F(1,28) = 9.192* ** ** *p = 0.005* ** ** *η^2^ = 0.108* **	*F(1,28) = 2.648* *p = 0.115* *η^2^ = 0.038*	** *F(1,28) = 4.566* ** ** *p = 0.041* ** ** *η^2^ = 0.069* **	F(1,28) = 4.9 *p* = 0.06 η^2^ = 0.069
ME embodiment	** *F(1,28) = 6.208* ** ** *p = 0.019* ** ** *η^2^ = 0.044* **	** *F(1,28) = 4.749* ** ** *p = 0.038* ** ** *η^2^ = 0.044* **	F < 1	F < 1
Interaction	** *F(1,28) = 9.429* ** ** *p = 0.005* ** ** *η^2^ = 0.082* **	F(1,28) = 2.223 *p* = 0.147 η^2^ = 0.016	F < 1	F < 1
GFP				
ME language	** *F(1,28) = 9.150* ** ** *p = 0.005* ** ** *η^2^ = 0.102* **	F < 1	F(1,28) = 2.974 *p* = 0.096 η^2^ = 0.041	** *F(1,28) = 6.678* ** ** *p = 0.015* ** ** *η^2^ = 0.09* **
ME embodiment	** *F(1,28) = 5.379* ** ** *p = 0.028* ** ** *η^2^ = 0.031* **	F < 1	F < 1	F < 1
Interaction	F(1,28) = 3.166 *p* = 0.086 η^2^ = 0.04	F(1,28) = 1.704 *p* = 0.202 η^2^ = 0.013	F < 1	F < 1

## Data Availability

Raw data and materials are accessible on the following public repositories, https://www.doi.org/10.5281/zenodo.4761369, accessed on 20 October 2024, through authorization of the authors, according to Swiss National Foundation rules.

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
