# Peer review of "Embodied Semantics: Early Simultaneous Motor Grounding in First and Second Languages"

_brainsci, 2024, doi:10.3390/brainsci14111056_

Round 1
Reviewer 1 Report
Comments and Suggestions for Authors
This paper investigates the embodiment of action verbs at different stages of lexical processing in the first language (L1) and a second language (L2). The study examines how action-related and non-action-related verbs in L1 and L2 recruit different neural mechanisms, using EEG to capture the temporal dynamics of embodiment. The central finding is that embodiment is observed at the lexical stage (N1 component) for both L1 and L2 but is not differentially embodied between the two languages. However, L2 processing is more effortful during semantic stages (P2/N400 components) than L1, which suggests that while embodiment mechanisms might be similar across languages, the overall processing in L2 requires more cognitive resources.
I found that the topic is worthwhile in psycholinguistics, the experiment design is sound, and the results are reliable. Therefore, I feel the paper is publishable on Brain Sciences with major revision.
One of the major issues with the manuscript is the lack of key references to foundational studies on embodied cognition, such as the work of Pulvermüller, Hauk, and related EEG research on verb processing. Without these critical citations, the theoretical framework appears weak and incomplete. The framework should explicitly explain the relationship between specific ERP components, such as N400 (associated with semantic processing), and the theory of embodied cognition.
The second problem lies in the absence of clear definitions for the lexical and semantic stages, as well as inconsistent usage of terms like early, middle, and late stages throughout the manuscript. These stages are crucial for interpreting the results and drawing final conclusions, and it is essential to provide consistent definitions and support them with authoritative references.
The third issue is that the authors do not clearly define the ERP components. The description of N1, P1, and N400 is unclear, and there is a lack of relevant literature citations. This undermines the authority and theoretical rigor of the discussion and conclusion sections.
Below, I would like to list my major comments to some issues I noticed in this manuscript.
1. Title: The current title lacks focus, is not concise, and uses too many prepositions. It is recommended to revise the title for clarity and precision.。
2. Abstract: “While this is well established for the mother tongue (L1), less is known about whether and how a language learned later in life through formal instruction (L2) is embodied.” The original wording is slightly awkward and could confuse the reader about what “this” refers to. It could be rephrased to: Although the embodiment of action-related language is well-established in the mother tongue (L1), less is known about the embodiment of a second language (L2) acquired later in life through formal instruction.
3. Abstract: The sentence “Results: We could show distinct effects of embodiment and language without interaction between embodiment and language...” is too lengthy and unclear. It should be broken down for clarity, and the key findings should be reworded in a more structured way. Additionally, the use of "could" is inappropriate.
4. Abstract: “Conclusions: Our results that processing words in L2 requires more effortful processing.” Suggestion: This sentence seems incomplete or grammatically incorrect. It could be refined for better readability.
5. P2: In the first paragraph, the statement “There are thus a number of evidences that the motor system is involved in language comprehension and that language is at least partially embodied. It remains a matter of debate whether embodiment affects earlier (lexical) and/or later (semantic) stages of language processing.” lacks sufficient citation support. To strengthen this argument, it is crucial to include references to relevant studies that provide empirical evidence for these claims.
6. P2: In the second paragraph, the statement “Most studies have investigated embodiment…” lacks citations. Relevant studies should be cited to support the claim and provide a stronger foundation for the discussion.
7. P2: In the second paragraph, “Given that L1 and L2 are acquired differently, one could assume that embodiment of L1 affects both early/word recognition and later/semantic stages…” What is the relationship between “word” and “lexical”? Please define this clearly and maintain consistency.
8. P3: In the second paragraph, the definitions and explanations of P1, N1, and P200 lack citations, reducing the authority of the discussion. Additionally, how do you account for cognitive processes that may already begin during the early stages of processing (e.g., phonological or orthographic recognition) and can overlap with sensory processing? Is there evidence suggesting that cognitive processes may not be as clearly separable from perceptual processes as indicated?
9. P3. The second paragraph suggests that P1 reflects visual analysis and N1 reflects early stages of visual word recognition and lexical processing. However, N1 is often considered part of the early sensory processing stage, which contrasts with the assertion that it reflects lexical processing. Can you provide more clarity on the role of N1 in lexical processing? How do you reconcile the idea that N1 is related to both sensory and lexical processing? Are there distinct subcomponents of N1 that can differentiate these processes?
10. P7: Please format Table 1 using the standard three-line table style. The same applies to other tables.
11. P7: Figure 2 is too blurry and unclear. Could you split it into separate parts for better clarity?
12. P7: In section 3.2, the analysis time windows for each component are not clearly specified. If possible, please indicate the time windows in the figure, and clarify in the text whether the time window selection was data-driven or based on previous literature. Additionally, explain the rationale behind this choice.
13. P8: In the second paragraph, there is a punctuation error with the “?” in the sentence: “neither the numbers of words known to the subjects nor the DIALANG scores correlated with any? ERP map strength (GFP) and explained variance (GEV).” Please correct this.
14. P9: In the first paragraph, “At the earliest stage of processing, we found a stronger P1 component in L2 than in L1 and for non-motor than motor verbs.” Please clearly specify the processing stage and ensure consistent terminology throughout the manuscript.
15. P9-p10. “Discussion” part should be arranged according to research questions.
16. P9-P10: In the discussion section, the study’s findings are not compared with other ERP studies on embodiment. Additionally, the reasons behind any observed differences are not thoroughly explored, which results in a lack of depth in the discussion. Please expand the discussion to include comparisons with relevant studies and provide a deeper analysis of potential causes for the differences.
Comments on the Quality of English LanguageSome expressions are not clearly defined as listed in my review report.
Author Response
This paper investigates the embodiment of action verbs at different stages of lexical processing in the first language (L1) and a second language (L2). The study examines how action-related and non-action-related verbs in L1 and L2 recruit different neural mechanisms, using EEG to capture the temporal dynamics of embodiment. The central finding is that embodiment is observed at the lexical stage (N1 component) for both L1 and L2 but is not differentially embodied between the two languages. However, L2 processing is more effortful during semantic stages (P2/N400 components) than L1, which suggests that while embodiment mechanisms might be similar across languages, the overall processing in L2 requires more cognitive resources.
I found that the topic is worthwhile in psycholinguistics, the experiment design is sound, and the results are reliable. Therefore, I feel the paper is publishable on Brain Sciences with major revision.
We would particularly like to thank the reviewer for his/her commitment and for the overall positive feedback and for his/her expert and the fruitful comments, which we tried to address as outlined below. We hope to have answered to the comments and to have improved the quality of the manuscript.
- One of the major issues with the manuscript is the lack of key references to foundational studies on embodied cognition, such as the work of Pulvermüller, Hauk, and related EEG research on verb processing. Without these critical citations, the theoretical framework appears weak and incomplete. The framework should explicitly explain the relationship between specific ERP components, such as N400 (associated with semantic processing), and the theory of embodied cognition.
We thank the reviewer for pointing out this important issue and we agree that the theoretical framework is weak. We did cite some important papers and experiments, Pulvermuller, F., et al., 2005, (Pulvermuller, F., et al., 2021, Tettamanti, M., 2005, Buccino et al 2016, Hauk, O. and F. Pulvermuller, 2004, but the explanatory part was effectively insufficient. We have now added a paragraph also insisting on theory, neuroimaging and ERP experiment. We have also brought a certain number of question concerning word representation and embodiment in L1 and L2 . Also, we discuss the meaning of the compelling evidence which argue for embodied representations in a second language.
The term “embodiment” refers to the grounding of cognition in systems involved in low level perceptual and action information processing. According to the experimental evidences of Pulvermüller, Hauk, Tettamanti and others, embodied theories of cognition claim that higher cognitive processing, including language, activates the same brain sen-sorimotor structures involved when experiencing the environment (Glenberg and Kaschak 2002, Hauk, Johnsrude et al. 2004, Pulvermuller, Hauk et al. 2005, Tettamanti, Buccino et al. 2005). One of the precursor of embodiments theories come from Neurophysiological investigations of noun and verb processing which provided distinct cortical topographies as biological counterparts of words (e.g. matched nouns and verbs), as early as 200 ms after word presentations (Pulvermuller, Lutzenberger et al. 1999). Subsequently, Hauk et al using event-related fMRI showed that action words referring to face, arm, or leg actions (such as lick, pick, or kick), presented in a passive reading task, differentially activated areas along the motor strip that either were directly adjacent to or overlapped with areas activated by actual movement of the tongue, fingers, or feet (Hauk, Johnsrude et al. 2004).
- The second problem lies in the absence of clear definitions for the lexical and semantic stages, as well as inconsistent usage of terms like early, middle, and late stages throughout the manuscript. These stages are crucial for interpreting the results and drawing final conclusions, and it is essential to provide consistent definitions and support them with authoritative references.
We understand the concern of the reviewer. We have now simplified the stages in early and late stages, since these two-Time windows are the most discussed in the literature. Moreover, we have more insisted on the components (P1, N1, P2, N400) and the processes (sensory, lexical and Semantic). This terminology has also been integrated in Fig 1. When referring to the literature, we have kept the original wording of the cited authors, particularly the delay in milliseconds in necessary. We have also added some precisions in the introductory presentation of Embodiment processing while reading.
…. (Papeo, Vallesi et al. 2009), the majority of evidence points toward early effects of embodi-ment within 200 ms after stimulus onset (Hauk and Pulvermuller 2004, Boulenger, Roy et al. 2006, Moseley, Pulvermuller et al. 2013, Vukovic, Feurra et al. 2017). It is however diffi-cult to disentangle the possible roles of the motor cortex and other lexical characteristics such as word frequency, physical orthographical, and lexical integration correlates taking place within 200 ms. (Pulvermuller, Shtyrov et al. 2009)
- The third issue is that the authors do not clearly define the ERP components. The description of N1, P1, and N400 is unclear, and there is a lack of relevant literature citations. This undermines the authority and theoretical rigor of the discussion and conclusion sections.
We thank the referee for this observation and suggestions. The paragraph on the introduction was not enough explicit and related to visual word recognition, thus rendering les clear the relation with embodiment. We have now tried to better define and refer the different components in the second part of the introduction, separating the early versus late components
The electro-cortical time course of word recognition has revealed several components linked with stages of word recognition, which have been classified as early and late com-ponents. As it will be outlined here, P1, N1 and P2 (or P200) are considered as early sensory and lexical components, while the late component N400 is considered a hallmark of semantic processing.
Early ERP components (up to ~150 – 200 ms after stimulus onset) are considered to reflect sensory and perceptual processes and vary with physical stimulus properties, In the case of word recognition and reading, the early components reflect the visual analysis (P1, generally occurring 100ms after stimulus presentation). The amplitude of the P1 component is particularly modulated by physical characteristics of words, such as font size, contrast, and their interaction, but also sometimes to emotional content(Schindler, Schettino et al. 2018). Then, the contextual automatic language process such as early stages of visual word recognition and lexical processing is reflected with the negative going potential component (N1, peaking at around 170 ms), (Eberhard-Moscicka, Jost et al. 2016). This Time Window is particularly sensitive to letter and word recognition (Schindler, Schettino et al. 2018). Particularly N1 peaked earlier to letters than pseudo letters (Schindler); such component is also sensitive to attentional charge.
An intermediate electric activity, starting with a P200p positive potential component, is generally integrated in the early component. The P200, a positive going potential component, has been related to orthographic and phonological pre-lexical processing well as to lexical access (Courson and Tremblay 2020).
The later endogenous components, whereas later ERPs (subsequent to ~200 – 250 ms after stimulus) are insensitive to physical variations of the stimulus and vary with respect to cognitive processes (Khateb, Pegna et al. 2010, Lee, Liu et al. 2012). Notably the N400, with a centro-parietal scalp distribution, reflects semantic judgment and integration during processing of the presented words, where amplitude increases negatively following semantic anomalies (Kutas and Hillyard 1980, Kutas and Hillyard 1984, Swaab, Brown et al. 1997, Kutas and Federmeier 2011). Originally, this event related potential (ERP) was seen as a signature of semantic incongruity, but since then other types of semantic tasks (e.g., semantic contextual integration, processing of single words out of context, tasks involving a concreteness effect) have shown the same robust signature (Pulvermuller, Shtyrov et al. 2009, Khateb, Pegna et al. 2010). For these reasons N400 is considered a hallmark of semantic processing. Bilingual subjects do recognize L2 words more slowly compared to L1, and the neurophysiological correlates of this target language difference are generally detected in the early rather than the late stage (Khateb, Pegna et al. 2016, de Leon Rodriguez, Mouthon et al. 2022).
Below, I would like to list my major comments to some issues I noticed in this manuscript.
- Title: The current title lacks focus, it is not concise and uses too many prepositions. It is recommended to revise the title for clarity and precision. 。
Thank you for this precious remark. We have reworded the title “ERP evidence of embodiment of action-verbs at lexical stages in L1 and L2” is now
Embodied Semantic: Early simultaneous motor grounding in first and second language
- Abstract: “While this is well established for the mother tongue (L1), less is known about whether and how a language learned later in life through formal instruction (L2) is embodied.” The original wording is slightly awkward and could confuse the reader about what “this” refers to. It could be rephrased to: Although the embodiment of action-related language is well-established in the mother tongue (L1), less is known about the embodiment of a second language (L2) acquired later in life through formal instruction.
Thank you for this suggestion, we have acknowledged your proposal and inserted it in the abstract.
Although the embodiment of action-related language is well-established in the mother tongue (L1), less is known about the embodiment of a second language (L2) acquired later in life through formal instruction.
- Abstract: The sentence “Results: We could show distinct effects of embodiment and language without interaction between embodiment and language...” is too lengthy and unclear. It should be broken down for clarity, and the key findings should be reworded in a more structured way. Additionally, the use of "could" is inappropriate.
Thank you for this remark. we have restructured the sentence and developed it for better focus and clarity
We could show distinct effects of embodiment and language without interaction between embodiment and language...”. words now
Reading action related verbs was associated higher ERP expression in the early phase of word recognition, particularly in the P1 time window, considered to reflect lexical process. Such effect was independent of the language context. There was no interaction between L1 and L2 context. Moreover, there were no effects of embodiment at the semantic stages of processing (N 400).
- Abstract: “Conclusions: Our results that processing words in L2 requires more effortful processing.” Suggestion: This sentence seems incomplete or grammatically incorrect. It could be refined for better readability.
To follow the research question, we have reworded the conclusion
Our results suggest that L1 and L2 are not embodied differently. The impact of embodiment was present and similar at the sensory and lexical (N1) stages of processing in both linguistic contexts. Moreover, our results suggest that processing words in L2 requires more effortful processing during the time window implicated in semantic process.
- P2: In the first paragraph, the statement “There are thus a number of evidence that the motor system is involved in language comprehension and that language is at least partially embodied. It remains a matter of debate whether embodiment affects earlier (lexical) and/or later (semantic) stages of language processing.” lacks sufficient citation support. To strengthen this argument, it is crucial to include references to relevant studies that provide empirical evidence for these claims.
We have added a certain number of references and consideration after the first sentence “There are thus a number of evidence that the motor system is involved in language comprehension and that language is at least partially embodied.”
Most fMRI studies on action language found activation in M1 and PM cortices in accordance with the effector to which the stimuli refer to. One of the first functional Magnetic Resonance (fMRI) studies in this respect is the one by Hauk et al. (2004), which showed how action verbs referring to the face, arm, or leg (e.g., to lick, pick, or kick) presented during a passive reading task activated the same areas appointed to the execution of the actions with the face, arm and leg respectively. Tettamanti et al. (2005) found a similar result with sentences involving actions during a listening task. Boulenger et al. (2009) found similar results while comparing idioms and literal sentences including arm- and leg-related action words.
It remains a matter of debate whether embodiment affects earlier (lexical) and/or later (semantic) stages of language processing
We have completer the references and explanations. We also have described some results and explicitly mentioned the authors to strengthen the argument
… . Some studies using single pulse Transcranial Magnetic Stimulation (spTMS), a technique which can influence a behavioral process in the range of tens of milliseconds (Pascu-al-Leone et al., 2000). Tomasino investigated semantic resonance (motor-to-language di-rectional effects) with this technique, an found embodiment effects within 200 ms, sug-gesting that it may take place at early stages of a word recognition(Tomasino, Fink et al. 2008). It is also possible to process semantics, e.g. by dissociating word categories (action vs. object) at about 150 ms and in modality-specific areas(Moseley, Pulvermuller et al. 2013). Hauk & Pulvermuller investigated early electrophysiological differences between matched semantic categories of action words in a passive reading task; significant differ-ences between subcategories of action words were present at approximately 220 msec(Hauk and Pulvermuller 2004). On the other hand, Buccino et al showed that lis-tening to action-related sentences with hand or foot effectors modulates the activity of the motor system about 500–700 ms after the beginning of the sentence suggesting that motor resonance of words may occur later, during semantic processing(Buccino, Riggio et al. 2005). In summary, while some data suggest that embodiment affects later semantic stages of processing (see also (Papeo, Vallesi et al. 2009), the majority of evidence points toward early effects of embodiment within 200 ms after stimulus onset (Hauk and Pulvermuller 2004, Boulenger, Roy et al. 2006, Moseley, Pulvermuller et al. 2013, Vukovic, Feurra et al. 2017). It is however difficult to disentangle the possible roles of the motor cortex and other lexical characteristics such as word frequency, physical orthographical, and lexical inte-gration correlates taking place within 200 ms (Pulvermuller, Shtyrov et al. 2009).
- P2: In the second paragraph, the
“Most studies have investigated embodiment…” lacks citations. Relevant studies should be cited to support the claim and provide a stronger foundation for the discussion.
The studies referenced in the following sentence “Most studies have investigated embodiment only in the mother tongue” are now cited in the prior paragraph. In order that the mentioned citations can also refer to this paragraph, we propose to adapt it in the following format.
The above-mentioned studies have investigated embodiment in participants’ mother tongue.
- P2: In the second paragraph, “Given that L1 and L2 are acquired differently, one could assume that embodiment of L1 affects both early/word recognition and later/semantic stages…” What is the relationship between “word” and “lexical”? Please define this clearly and maintain consistency.
We thank the reviewer for this precise remark. We used here Word recognition, as the moment where the letter string is recognized as a word. We refer here to the lexical stage, which is generally considered to happen earlier than the semantic process itself. This has now been detailed in the former paragraph_ line 83 to 119
(“Given that L1 and L2 are acquired differently, one could assume that embodiment of L1 affects both early/word recognition and later/semantic stages…” is now
Given that L1 and L2 are acquired differently, one could assume that embodiment of L1 affects both early/lexical and later/semantic stages…”
- P3: In the second paragraph, the definitions and explanations of P1, N1, and P200 lack citations, reducing the authority of the discussion. Additionally, how do you account for cognitive processes that may already begin during the early stages of processing (e.g., phonological or orthographic recognition) and can overlap with sensory processing? Is there evidence suggesting that cognitive processes may not be as clearly separable from perceptual processes as indicated?
We thank the referee for this observation and suggestions, which reflect the general concern (particularly the third point) mentioned at the beginning of the review. As mentioned before, we have developed the time course of word recognition, splitting the early and late components, the paragraph starts line 83.
The electro-cortical time course of word recognition has revealed several components linked with stages of word recognition, which have been classified as early and late components. As it will be outlined here, P1, N1 and P2 (or P200) are considered as early sensory and lexical components, while the late component N400 is considered a hallmark of semantic processing.
Early ERP components (from 50 ms up to ~150 – 200 ms after stimulus onset) are considered to reflect sensory and perceptual processes and vary with physical stimulus properties, In the case of word recognition and reading, the early components reflect the visual analysis (P1, generally occurring 100ms after stimulus presentation). The amplitude of the P1 component is particularly modulated by physical characteristics of words, such as font size, contrast, and their interaction, but also sometimes to emotional content(Schindler, Schettino et al. 2018). Then, the contextual automatic language process such as early stages of visual word recognition and lexical processing is reflected with the negative going potential component (N1, peaking at around 170 ms), (Eberhard-Moscicka, Jost et al. 2016). This Time Window is particularly sensitive to letter and word recognition(Schindler, Schettino et al. 2018). Particularly N1 peaked earlier to letters than pseudo letters; (Schindler, Schettino et al. 2018) and such component is also sensitive to attentional charge. In addition, while the N1 component is generally linked to early sensory and prelexical processing, it has also been associated with early lexical activation (Mahé et al., 2013; Hauk et al., 2012). Different studies indicate that the N1 comprises multiple subcomponents (Korinth, Sommer, & Breznitz, 2013; Eberhard-Moscicka et al., 2016), with its later phase showing sensitivity to lexical processing (Eberhard-Moscicka et al., 2016). According to Eberhard-Moscicka, adult speakers of English as a foreign language exhibited print tuning in the early N1 andlexicality effects in the late N1, indicating that sensitivity to print and lexicality unfolds differently throughout the N1 segment in adults (Eberhard-Moscicka, Jost et al. 2016). An intermediate electric activity, starting with a P200, a positive going potential component, has been related to orthographic and phonological pre-lexical processing well as to lexical access (Courson and Tremblay 2020).
The later endogenous components, whereas later ERPs (subsequent to ~200 – 250 ms after stimulus) are insensitive to physical variations of the stimulus and vary with respect to cognitive processes (Khateb, Pegna et al. 2010, Lee, Liu et al. 2012). Notably the N400, with a centro-parietal scalp distribution, reflects semantic judgment and integration during processing of the presented words, where amplitude increases negatively following semantic anomalies (Kutas and Hillyard 1980, Kutas and Hillyard 1984, Swaab, Brown et al. 1997, Kutas and Federmeier 2011). Originally, this event related potential (ERP) was seen as a signature of semantic incongruity, but since then other types of semantic tasks (e.g., semantic contextual integration, processing of single words out of context, tasks involving a concreteness effect) have shown the same robust signature (Pulvermuller, Shtyrov et al. 2009, Khateb, Pegna et al. 2010). For these reasons N400 is considered a hallmark of semantic processing. Bilingual subjects do recognize L2 words more slowly compared to L1, and the neurophysiological correlates of this target language difference are generally detected in the early rather than the late stage (Khateb, Pegna et al. 2016, de Leon Rodriguez, Mouthon et al. 2022).
- P3. The second paragraph suggests that P1 reflects visual analysis and N1 reflects early stages of visual word recognition and lexical processing. However, N1 is often considered part of the early sensory processing stage, which contrasts with the assertion that it reflects lexical processing. Can you provide more clarity on the role of N1 in lexical processing? How do you reconcile the idea that N1 is related to both sensory and lexical processing? Are there distinct subcomponents of N1 that can differentiate these processes?
We agree with the reviewer that the description of the N1 is too short, given also the fact that it is a important component in our hypothesis and our results. We have completed the description of both components and integrated it in the introduction (integrated also in point 9)
- In addition, while the N1 component is generally linked to early sensory and prelexical processing, it has also been associated with early lexical activation (Mahé et al., 2013; Hauk et al., 2012). Different studies indicate that the N1 comprises multiple subcomponents (Korinth, Sommer, & Breznitz, 2013; Eberhard-Moscicka et al., 2016), with its later phase showing sensitivity to lexical processing (Eberhard-Moscicka et al., 2016). According to Eberhard-Moscicka, adult speakers of English as a foreign language exhibited print tuning in the early N1 andlexicality effects in the late N1, indicating that sensitivity to print and lexicality unfolds differently throughout the N1 segment in adults (Eberhard-Moscicka, Jost et al. 2016).
An intermediate electric activity, starting with a P200, a positive going potential component, has been related to orthographic and phonological pre-lexical processing well as to lexical access (Courson and Tremblay 2020).
- P7: Please format Table 1 using the standard three-line table style. The same applies to other tables.
This has been done, the table is now formatted with the 3-lines table style
Table 1. L2-proficiency scores (mean/ Standard deviation) assessed by the DIALANG and by the translation recognition task for the L2 verbs used in the EEG experiment.
|
all subjects |
L1 French |
L1 German |
DIALANG L2 scores |
445.3 / 252.6 |
377.3 / 246.8 |
512.1 / 228.4 |
translation recognition in % |
|
|
|
all words |
70.45 / 18.9 |
62.2 / 14.8 |
79.3 / 18.1 |
motor verbs |
70.48 / 19.9 |
64.0 / 17.7 |
77.4 / 20.5 |
non- motor verbs |
73.86 / 18.3 |
67.0 / 16.3 |
81.1 / 17.9 |
|
|
|
|
- P7: Figure 2 is too blurry and unclear. Could you split it into separate parts for better clarity?
After reviewing Figure 2, we agree that Fig 2 was not clear. Particularly, the important data are in the left side window, while the right-side window is an example of analysis and brought more confusion than else. Moreover, since we focus on the motor non motor results and the interaction, we have decided to present only the significant motor non motor results, since the L1/L2 results are already in the table. We hope in this way the data are more understandable and allow also to focus the discussion. Here is the new figure
Figure 2 Results of the spatiotemporal segmentation procedure for the analysed words. A. Illustration of the computed segmentation for each experimental condition (amplitude of plots is the Global Field Power [GFP]). B. Topographies of the microstates isolated. C. Graphical illustration when Motor relatedness factor is significant for the global explained variance (GEV) and GFP (details in table 2). White dots = motor related verbs (M), black dots = non-motor related verbs (nM), *=0.05>p>0.01, **=p<0.01. L1 = mother tongue, L2 = second language learn at later stage. Map 2, which correspond to the largest part of the P100 time windows, extends between 100 and 150 ms after stimulus presentations. Map 4. Which corresponds to N1 extends between around 150 and 300 ms after stimulus presentations.
- P7: In section 3.2, the analysis time windows for each component are not clearly specified. If possible, please indicate the time windows in the figure, and clarify in the text whether the time window selection was data-driven or based on previous literature. Additionally, explain the rationale behind this choice.
Thank you very much for this very important point. As mentioned in the method, the time window selection is data driven. This is referred to in the methods
“ In order to identify the dominant topographies, we used a spatial atomize agglomerate hierarchical cluster analysis (AAHC, (Murray, Brunet et al. 2008, Britz and Michel 2010, Britz and Pitts 2011) …… We computed 20 different solutions of the AAHC analysis and identified the optimal solution by the convergence of a combination of multiple criteria (Brechet, Brunet et al. 2019) that attempt to maximize the explained variance with a minimum number of clusters. Because ERP map topographies of < 10 ms are physiologically implausible, we applied a temporal constraint criterion of 10 ms.”
At the beginning of the paragraph, we re-insisted on the hypothesis and the methodological choice
According to our hypotheses the periods of interest should corresponding P1, N1 and P2 components during the early processing phase of word recognition and N400 component for the later, “semantic” phase. To determine them without bias, a data driven approach was applied.
And we also specified the time windows extension in the results and in the figure
The results and interactions are summarized in Figure 2 and Table 2. The illustration A of figure 1 shows the different time windows of the Maps identified through cluster analysis. Map 2, corresponding to the quasi totality of P1 runs from 100 to 150 ms. Maps 4, corresponding to N1, runs from 150 to 300 ms. The maps number 6, 7 and 8, corresponding to the N400 time window, extends essentially from 400 ms to 500 ms.
- P8: In the second paragraph, there is a punctuation error with the “?” in the sentence: “neither the numbers of words known to the subjects nor the DIALANG scores correlated with any? ERP map strength (GFP) and explained variance (GEV).” Please correct this.
Thank you and sorry for the mistake. This has been corrected
- P9: In the first paragraph, “At the earliest stage of processing, we found a stronger P1 component in L2 than in L1 and for non-motor than motor verbs.” Please clearly specify the processing stage and ensure consistent terminology throughout the manuscript.
Thank you for this important point. Given referee’s remark, we have now better defined the processing stages, as mentioned before. So P1 and N1 are better defined, in our impression. Particularly, we have now added this information at the beginning of results paragraph for ERP (see point 13)
The results and interactions are summarized in Figure 2 and Table 2. The illustration A of figure 1 shows the different time windows of the Maps identified through cluster analysis. Map 2, corresponding to the quasi totality of P1 runs from 100 to 150 ms. Maps 4, corresponding to N1, runs from 150 to 300 ms. The maps number 6, 7 and 8, corresponding to the N400 time window, extends essentially from 400 ms to 500 ms.
We have also simplified the figure to focus on motor/ non motor question (our main question) and let the other effects and interactions in the table. We hope that this is now clearer.
- P9-p10. “Discussion” part should be arranged according to research questions.
We agree that Discussion can be reorganized and we decided to summarize the research question and the main results in the first paragraph of the discussion.
Our main research questions question was i) to replicate electrophysiological corre-lates of word embodiment in silent reading task and ii) to compare the time course of embodiment of action verbs in L1 and L2 and eventually detect differences in time and amount of brain activity through global field power and global explained variance analysis. The main results were the following: i) grounding of action words occurs in early phase of word recognition at the level of P1 map (corresponding to 100 ms after word presentation) and N1 map (corresponding to 150 to 300 ms after word presenta-tion ii) This process is similar in time for both languages, and associated to larger ac-tivity in L2 at the level of the P1 time window. Iii) we found an early interaction in EEG activity between motor /non-motor words andL1/L2 context, with a stronger P1 component in L2 than in L1 and for non-motor than motor verbs.
Then we have reorganized the discussion, leaving the summary of the results at the beginning, and focussing on comparing with other ERP and and fmri data. We hope in this way to have brought a more solid discussion.
With the high temporal resolution of ERPs, we show here that embodiment within this task exclusively affects lexical stages of processing and equally so for L1 and L2, and that the more effortful processing of L2 selectively affects here word recognition and semantic stage of processing. This early impact of embodiment in the process of word recognition (P1 and N1) confirmed other ERP studies. Vukovic and Shtyrov (2014), for example, examined mu-rhythm desynchronization as an index of motor cortex activity in response to L1 and L2 abstract and action prime-probe verb pairs. They found that cortical motor activation was present in both L1 and L2 around 150 ms post-stimulus. Studies on novel word learning showed also early motor grounding effects (e.g., Vukovic & Shtyrov, 2019, see Kogan, Muñoz, et al., 2020, for a review) suggesting that such effects are also early when learning a second language. Interest-ingly, Xue et al. (2015) presenting to L1-Ch L2-English participants with high (e.g., crumb) and low (e.g.,lace) body-object interaction (BOI) English words obtained a slightly later embodiment effect. These words were imbedded in high (e.g., you brush the small sticky crumb) and low (e.g., you wear a string of cotton lace) sensorimotor contexts. Highly proficient L2-English participants judged sentence acceptability while ERPs time-locked to the onset of the high vs. low BOI words in rich and poor context were recorded. The results showed a marginal sensorimotor context effect reflected in ERP differences in later time-windows, P2 and N400 components. It is possible that impact on later time windows is due to the type of task, integrating also syntactic and sentence related information. our findings are in line with evidence from fMRI that shows no differences in embodiment but stronger recruitment of language areas in L2 than in L1 (De Grauwe, Willems et al. 2014, Tian, Chen et al. 2020, Zhang, Yang et al. 2020) indicating that L1 and L2 are embodied to comparable degrees. Due to the slow time-course of the BOLD response, these fMRI studies cannot distinguish whether em-bodiment affects lexical or semantic stages of processing, or whether L1 and L2 might be embodied at different stages of processing..
Our results are in contrast with those from behavioral and neurophysiological studies that found a stronger embodiment of L1 than L2. Behavioral measures (reac-tion times, accuracy rates) only reflect the end-product of a cognitive process with a certain delay and they cannot reveal which stage of processing contributes to the effect. For the case of embodiment, e.g. reaction times cannot distinguish whether em-bodiment affects early or later stages of processing or whether embodiment affects different stages of processing in L1 and in L2. If the strength of embodiment was re-flected in this component, it should be more strongly pronounced in the motor than in the non-motor condition (de Zubicaray, McMahon et al. 2021), which is not the case. One potential explanation for the stronger P1 (early visual potential) component in L2 may reflect interlanguage differences in ortho-phonological processing (de Zubicaray, Arciuli et al. 2023) with possible decreased typicity between perception and meaning in L2. Another explanation for such stronger GFP in L2 at 100 ms could be explained by larger co-activation of the embodiment effect in the early stages of word recognition in L2 (e.g., Kroll, Gullifer, & Rossi, 2013; Dijkstra & van Heuven, 2002). Access to L2 representations would require mediation via L1, especially in case of low L2 proficien-cy. This entails a later sensorimotor involvement when L2 proficiency is low compared to when it is high or compared to L1. Such differences in the degree of L2 embodiment would also be in line with the Dijkstra’s BIA+ model. Birba et al. (2020) investigated L1 and L2 embodiment in action and non-action-laden narratives, and, despite showing motor-related connectivity in L2, they could not show differential activation between the two types of texts in L2. Overall, the first conclu-sions rising from this new emerging topic of research point toward early embodiment of a late acquired L2 as similar to L1 Embodiment, but integrating possibly other as-pect of action–language interaction deserving to be better delineated in its nuances
- P9-P10: In the discussion section, the study’s findings are not compared with other ERP studies on embodiment. Additionally, the reasons behind any observed differences are not thoroughly explored, which results in a lack of depth in the discussion. Please expand the discussion to include comparisons with relevant studies and provide a deeper analysis of potential causes for the differences.
Thank you for this important point. See our answer on point 15.
Reviewer 2 Report
Comments and Suggestions for Authors
Review
“ERP evidence of embodiment of action-verbs at lexical stages in L1 and L2”
General assessment:
- This study investigates differences in how L1 and L2 are embodied during language processing. Late French-German and German-French bilinguals silently read action-related and non-action-related verbs, and neurophysiological data were recorded to capture processing dynamics. The study controls for language acquisition sequence by collapsing across languages. The results are interpreted to demonstrate that embodiment effects emerge exclusively in early lexical stages, with action-related verbs eliciting stronger early brain responses, showing identical patterns across L1 and L2. This suggests that embodiment mechanisms are consistent across both language systems. However, later semantic stages show significant divergence: L2 requires greater cognitive resources for integration, as evidenced by stronger neural responses at later processing stages. These results indicate that while both L1 and L2 share similar embodiment processes at early stages, L2 requires more effortful processing during semantic interpretation due to reduced automaticity and proficiency. The study concludes that while embodiment is similar in L1 and L2, processing in L2 is more cognitively demanding at later stages of linguistic processing.
- In general, I find the topic of the paper highly compelling, particularly given the limited attention it has received in the literature concerning the embodiment of languages other than L1. It is also highly relevant to the special issue on The Relationship between Language Processing and Cognitive Development. The paper's objective is clearly articulated, the methodology is robust, and the authors effectively engage with relevant literature and previous research.
- For these reasons, I support the publication of this paper. However, I believe it is not ready for publication in its current form. Several issues need to be addressed before submitting the final version (see below).
My overall recommendation is:
ACCEPT AFTER MINOR REVISION
Content:
- General remark: There is an unmistakable detail that I find particularly concerning in this paper—namely, the authors assume certain concepts without adequately clarifying them. For instance, the use of the term "L2" implicitly tied to the identification of a foreign or second language is plausible, but this needs to be more explicitly defined. "L2" is used in a variety of contexts and interpretations within the literature. Furthermore, the authors refer to "late acquisition" as formal (declarative) learning occurring after the age of 5 to 7. On what basis is this threshold established? Why is this specific cut-off used to define the boundary between early and late acquisition? What evidence supports this distinction? These points require further elaboration, as researchers in the field of L2/L3 acquisition from other theoretical perspectives may find this framing unclear and potentially confusing.
- p. 2: …the acquisition of semantic concepts is assumed to be a direct consequence of that interaction with the environment > By whom? And on the basis of what evidence?
- p. 3: (n = 31, 2 males, mean age: 27 years, SD: 6) and German (n = 41, 5 males, mean age: 25 years, SD: 5) > Please provide the EXACT mean ages.
- p. 3: Stimuli were action-verbs referring to motor actions executed by the hands (“grasp”) and non-motor related verbs (“guess”) in French and German. > Are “grasp” and “guess” examples or the only verbs utilized? Please also provide the actual verbs used in the study (the French and German ones).
- Description of the test subjects in general + p. 7 and following pages (Section 3): A detail that is discussed too superficially is the thorough characterization of the speakers, including the context and type of L2 acquisition they experienced. For example, are the German speakers from Germany, Austria or Switzerland (or do you have a mixed situation there)? If they are from Switzerland, their exposure to French might differ significantly from that of L2 learners in Germany. Similarly, are the French speakers from France, Switzerland, or another country/region? This distinction could also lead to notable differences in their language acquisition experience. Where did they learn the L2 – at school? privately? etc. …
- A question that arises after reading the paper is whether the level of embodiment and/or cognitive demand might correlate with the degree of linguistic kinship between the languages compared or with how a language conceptualizes certain types of movement. German and French, while excellent candidates for such a study from a comparative-linguistic standpoint due to the different lexical roots of most tested elements, leave me questioning whether the results of this study can be generalized. Specifically, can it be claimed that L1 and L2 embodiment are similar cross-linguistically, but L2 embodiment is more cognitively demanding cross-linguistically as well?
- Could the authors provide a comprehensive list of the stimuli used, such as the 200 verbs, for further analysis?
Formal aspects:
- General remark: Contrary to my prior experience as a reviewer for this journal, the lines in the document are not numbered, which makes it somewhat difficult to reference specific sections in the text.
- p. 1: s (symbol grounding problem, [2]…. > The bracket is not closed.
- p. 2 and elsewhere: Some acronyms are introduced and then never mentioned again. E.g.: Physiological measures with high temporal resolution such as electro- (EEG) and magnetoencephalography (MEG) and to a lesser degree transcranial magnetic stimulation (TMS) > I suggest the authors use only the full form in such cases.
- p. 2: While there is some data suggest > suggesting
- p. 2: (first language/ L1) > (first language / L1) or (first language/L1)
- p. 3: whereas later ERPs > It appears there may be an extra space before "ERPs" (potentially caused by the PDF conversion). If so, please remove the unnecessary space.
- p. 5: The image resolution of Figure 1 is quite low. Could you please provide a higher-quality version for better clarity?
Comments on the Quality of English Language(See above)
Author Response
General assessment:
- This study investigates differences in how L1 and L2 are embodied during language processing. Late French-German and German-French bilinguals silently read action-related and non-action-related verbs, and neurophysiological data were recorded to capture processing dynamics. The study controls for language acquisition sequence by collapsing across languages. The results are interpreted to demonstrate that embodiment effects emerge exclusively in early lexical stages, with action-related verbs eliciting stronger early brain responses, showing identical patterns across L1 and L2. This suggests that embodiment mechanisms are consistent across both language systems. However, later semantic stages show significant divergence: L2 requires greater cognitive resources for integration, as evidenced by stronger neural responses at later processing stages. These results indicate that while both L1 and L2 share similar embodiment processes at early stages, L2 requires more effortful processing during semantic interpretation due to reduced automaticity and proficiency. The study concludes that while embodiment is similar in L1 and L2, processing in L2 is more cognitively demanding at later stages of linguistic processing.
- In general, I find the topic of the paper highly compelling, particularly given the limited attention it has received in the literature concerning the embodiment of languages other than L1. It is also highly relevant to the special issue on The Relationship between Language Processing and Cognitive Development. The paper's objective is clearly articulated, the methodology is robust, and the authors effectively engage with relevant literature and previous research.
- For these reasons, I support the publication of this paper. However, I believe it is not ready for publication in its current form. Several issues need to be addressed before submitting the final version (see below).
My overall recommendation is:
ACCEPT AFTER MINOR REVISION
We are very grateful to the reviewer for his/her commitment and for the encouraging and positive feedback and for his/her expert and the fruitful comments, which we tried to address as outlined below. We hope to have answered to the comments and to have improved the quality of the manuscript.
.
Content:
- General remark: There is an unmistakable detail that I find particularly concerning in this paper—namely, the authors assume certain concepts without adequately clarifying them. For instance, the use of the term "L2" implicitly tied to the identification of a foreign or second language is plausible, but this needs to be more explicitly defined. "L2" is used in a variety of contexts and interpretations within the literature. Furthermore, the authors refer to "late acquisition" as formal (declarative) learning occurring after the age of 5 to 7. On what basis is this threshold established? Why is this specific cut-off used to define the boundary between early and late acquisition? What evidence supports this distinction? These points require further elaboration, as researchers in the field of L2/L3 acquisition from other theoretical perspectives may find this framing unclear and potentially confusing.
We thank the reviewer for this very important theoretical question. As mentioned by the reviewer there are different context and interpretations on this issue. We refer here to the to the age in which participants have started to learn the additional second language. We rely on the work of Pavlenko, Dulchig and Abutalebi, which we cite now in a specific paragraph
Generally, the term Age of Acquisition (AoA) refers to the age at which one begins to learn an additional language, which, may or may not coincide with the age of arrival in the context of that language (Pavlenko, 2012). AoA has been used to classify bilinguals into simultaneous (AoA=from birth), early/childhood (AoA=prior to age 7 or 12, depending on the studies) and late/adults bilinguals (AoA=after the age of 7 or 12, or post-puberty) (e.g., Pavlenko, 2012). In our research, we classify as late bilinguals the participants who learn L2 after 7 years of age (Dudschig et al., 2014). According to experimental data and theoretical models, representations of concepts in the first (L1) and second language (L2 partially overlap particularly in early bilinguals (Dudschig et al., 2014; Jeong et al., 2021, as notably evidenced by neuroimaging studies (Abutalebi et al., 2008). However, in late bilinguals, L2 vocabulary is often acquired in late childhood and typically in a classroom context (particularly when learnt in secondary school), which is mostly based on explicit memory. In Switzerland, German and French are the most important national languages and are formally learned respectively by L1 French and German speaking children at school in the 5th grade (8 years old). There is no immersive learning at this stage. Therefore, the question arises if sensorimotor experience is involved in late L2 learning concepts. Based on this assumption, we might suggest semantic representations in L2 to be less embodied than in L1 (Dudschig et al., 2014).
Dudschig, C., de la Vega, I., & Kaup, B. (2014). Embodiment and second-language: Automatic activation of motor responses during processing spatially associated L2 words and emotion L2 words in a vertical Stroop paradigm. Brain and Language, 132, 14–21. https://doi.org/10.1016/j.bandl.2014.02.002
Abutalebi, J. (2008). Neural aspects of second language representation and language control. Acta Psychol. 128, 466–478. doi: 10.1016/j.actpsy.2008.03.014
Pavlenko, A. (2012). Affective processing in bilingual speakers: Disembodied cognition? International Journal of Psychology, 47, 405–428. https://doi.
- p. 2: …the acquisition of semantic concepts is assumed to be a direct consequence of that interaction with the environment > By whom? And on the basis of what evidence?
This point is highly important and lack of theoretical references for embodied semantic has also been pointed out by Reviewer 1. We have then added in the text that Embodied cognition postulates that higher cognitive processing, including language processing, activates similar neural sensorimotor structures involved when experiencing the environment. The text and references concerning this particular point are now added in the introduction
The term “embodiment” refers to the grounding of cognition in systems involved in low level perceptual and action information processing. According to the experimental evidences of Pulvermüller, Hauk, Tettamanti and others, embodied theories of cognition claim that higher cognitive processing, including language, activates the same brain sen-sorimotor structures involved when experiencing the environment (Glenberg and Kaschak 2002, Hauk, Johnsrude et al. 2004, Pulvermuller, Hauk et al. 2005, Tettamanti, Buccino et al. 2005). One of the precursor of embodiments theories come from Neurophysiological investigations of noun and verb processing which provided distinct cortical topographies as biological counterparts of words (e.g. matched nouns and verbs), as early as 200 ms after word presentations (Pulvermuller, Lutzenberger et al. 1999). Subsequently, Hauk et al using event-related fMRI showed that action words referring to face, arm, or leg actions (such as lick, pick, or kick), presented in a passive reading task, differentially activated areas along the motor strip that either were directly adjacent to or overlapped with areas activated by actual movement of the tongue, fingers, or feet (Hauk, Johnsrude et al. 2004).
- p. 3: (n = 31, 2 males, mean age: 27 years, SD: 6) and German (n = 41, 5 males, mean age: 25 years, SD: 5) > Please provide the EXACT mean ages.
In our Redcap databank we have now the global data, so that we have inserted now the mean age of the global population who underwent the test
A total of 361 French and German verbs were randomly split into two lists that were administered to naïve University students (), mean age: 21.55 years, SD: 2,44), native speakers of French (n = 31, 2 males), and German (n = 41, 5 males respectively,
- p. 3: Stimuli were action-verbs referring to motor actions executed by the hands (“grasp”) and non-motor related verbs (“guess”) in French and German. > Are “grasp” and “guess” examples or the only verbs utilized? Please also provide the actual verbs used in the study (the French and German ones).
Thank you for this remark, which let us think that we were not clear enough. On fact the initial number of stimuli is 361, as mentioned under in the text. Two surveys were conducted prior to the study to create the stimulus material. Grasp and Guess are example of utilized verbs. The text is now modified Sentence
The stimuli used for the experimental paradigm were 200 verbs, composed of action-verbs referring to motor actions executed by the hands (for example “grasp”) and non-motor verbs (for example “guess”) in French and German. Two pre-tests were conducted …
- Description of the test subjects in general + p. 7 and following pages (Section 3): A detail that is discussed too superficially is the thorough characterization of the speakers, including the context and type of L2 acquisition they experienced. For example, are the German speakers from Germany, Austria or Switzerland (or do you have a mixed situation there)? If they are from Switzerland, their exposure to French might differ significantly from that of L2 learners in Germany. Similarly, are the French speakers from France, Switzerland, or another country/region? This distinction could also lead to notable differences in their language acquisition experience. Where did they learn the L2 – at school? privately? etc. …
The reviewer is quite right concerning the lack of precision in our description. Actually, we controlled precisely that the included participants were early German and French Monolingual, late French and German L2 speakers, and had done their schooling in Switzerland in order to have a homogeneous background. We are sorry that we were not precise enough. We added information in the text now
normal or corrected-to-normal visual acuity and were right-handed. They had either a high school diploma (Matura) or a university degree (bachelor or master), which is a minim um of 13 years of education. Fribourg is a bilingual University, and participants had lectures and seminar in French or German, but some courses were also in English. Our experimental group included native German speaking and native French speaking students. Inclusion criteria for the study was that L1 German speaking participants had done primary school in German part of Switzerland and were exposed only to Swiss German and/or German before the age of 7; and the L1 French group was only exposed to standard French before age of 7. According to Swiss schooling rules, L2 language was French for the German group and standard German for the French group. Immersive second language learning program in the other language before the end of primary school as well as early bilingualism were exclusion criteria. Simulta-neous or later third language was generally English in both groups.
- A question that arises after reading the paper is whether the level of embodiment and/or cognitive demand might correlate with the degree of linguistic kinship between the languages compared or with how a language conceptualizes certain types of movement. German and French, while excellent candidates for such a study from a comparative-linguistic standpoint due to the different lexical roots of most tested elements, leave me questioning whether the results of this study can be generalized. Specifically, can it be claimed that L1 and L2 embodiment are similar cross-linguistically, but L2 embodiment is more cognitively demanding cross-linguistically as well?
We thank Reviewer 2 for bringing this issue to attention. Indeed, linguistic distance between the L1 and L2 has been highlighted in past research as having a key role in cross-language decoding (e.g., Jeong et al., 2007, Kim et al., 2016), as well as general bilingual language processing (Abutalebi et al., 2015; Ghazi-Saidi & Ansaldo, 2017). While we agree with the Reviewer that such a question has substantial theoretical grounds and would benefit further research, there have neither been any systematic investigation into whether such linguistic distance, including characteristics such as overlaps in vocabulary, grammar, or script, may predict the extent of L2 embodiment, nor has there been any compelling evidence to support such a claim. To the best of our knowledge, this issue was to a certain extent examined in a study by Ahlberg et al. (2017), which assessed L1 and L2 embodiment effects by comparing native German speakers with non-native German speakers whose L1 either similarly or distinctly encodes spatial relations. Although embodiment effects were found for both non-native speaker groups, these effects were not affected by the linguistic distance between their L1 and L2. These results speak to the fact that although L2 embodiment may in some respects show similarities with L1, the extent of L2 embodiment may not necessarily correlate with the degree of overlap between the languages compared. However, we would argue that more research is necessary to examine such an issue.
Abutalebi, J., Canini, M., Della Rosa, P. A., Green, D. W., and Weekes, B. S. (2015). The neuroprotective effects of bilingualism upon the inferior parietal lobule: a structural neuroimaging study in aging Chinese bilinguals. Journal of Neurolinguistics. 33, 3–13. doi: 10.1016/j.jneuroling.2014.09.008
Ahlberg, D. K., Bischoff, H., Kaup, B., Bryant, D., and Strozyk, J. V. (2017). Grounded cognition: comparing Language × Space interactions in first language and second language. Applied Psycholinguistics. 39, 437–459. doi: 10.1017/ S014271641700042X
Ghazi-Saidi, L., & Ansaldo, A. I. (2017). The neural correlates of semantic and phonological transfer effects: language distance matters. Bilingualism Language and Cognition. 20, 1080–1094. doi: 10.1017/S136672891600064X
We added a focused summary of our response among the limitations.
Finally, we did not quantified the linguistic distance, including characteristics such as overlaps in vocabulary, grammar, or script, which could modulate the extent of L2 embodiment, To the best of our knowledge, this issue was to a certain extent examined in a study by Ahlberg et al. (2017), which assessed L1 and L2 embodiment effects by comparing native German speakers with non-native German speakers whose L1 either similarly or distinctly encodes spatial relations. Although embodiment effects were found for both non-native speaker groups, these effects were not affected by the linguistic distance between their L1 and L2[71]. These results speak to the fact that the extent of L2 embodiment may not necessarily correlate with the degree of overlap between the languages compared. However, we would argue that more research is necessary to examine such an issue.
Jeong, H., Sugiura, M., Sassa, Y., Haji, T., Usui, N., Taira, M., Horie, K., Sato, S., & Kawashima, R. (2007). Effect of syntactic similarity on cortical activation during second language processing: a comparison of English and Japanese among native Korean trilinguals. Human Brain Mapping, 28(3), 194–204. https://doi.org/10.1002/hbm.20269
Kim, S. Y., Qi, T., Feng, X., Ding, G., Liu, L., & Cao, F. (2016). How does language distance between L1 and L2 affect the L2 brain network? An fMRI study of Korean-Chinese-English trilinguals. NeuroImage, 129, 25–39. https://doi.org/10.1016/j.neuroimage.2015.11.068
- Could the authors provide a comprehensive list of the stimuli used, such as the 200 verbs, for further analysis?
This is indeed an important point. We have now provided the list on an pdf document as supplementary material. We hope that the information is sufficiently useful.
Formal aspects:
- General remark: Contrary to my prior experience as a reviewer for this journal, the lines in the document are not numbered, which makes it somewhat difficult to reference specific sections in the text.
Thank you for this remark. This has been now changed
- p. 1: s (symbol grounding problem, [2]…. > The bracket is not closed.
Done, thank you very much
- p. 2 and elsewhere: Some acronyms are introduced and then never mentioned again. E.g.: Physiological measures with high temporal resolution such as electro- (EEG) and magnetoencephalography (MEG) and to a lesser degree transcranial magnetic stimulation (TMS) > I suggest the authors use only the full form in such cases.
We have effectively erased MEG and TMS acronym. We kept only EEG acronym, since it is now more present in the text. We hope so to improve readability
- p. 2: While there is some data suggest > suggesting
We have added a certain number of references and introduction on suggestion of the first reviewer. So, the sentence is now:
In summary, while some data suggest that embodiment….
- p. 2: (first language/ L1) > (first language / L1) or (first language/L1)
Modified for first language / L1
- p. 3: whereas later ERPs > It appears there may be an extra space before "ERPs" (potentially caused by the PDF conversion). If so, please remove the unnecessary space.
This has been deleted
- p. 5: The image resolution of Figure 1 is quite low. Could you please provide a higher-quality version for better clarity?
Thank you for these important remarks We agree with the insufficient quality of the Figure. Also, the important data are in the left side window, while the right-side window is an example of analysis and brought more confusion than else. Moreover, since we focus on the motor non motor results and the interaction, we have decided to present only the significant motor non motor results, since the L1/L2 results are already in the table. We hope in this way the data are more understandable and allow also to focus the discussion. The new figure is simpler and of better quality.
Figure 2 Results of the spatiotemporal segmentation procedure for the analysed words. A. Illustration of the computed segmentation for each experimental condition (amplitude of plots is the Global Field Power [GFP]). B. Topographies of the microstates isolated. C. Graphical illustration when Motor relatedness factor is significant for the global explained variance (GEV) and GFP (details in table 2). White dots = motor related verbs (M), black dots = non-motor related verbs (nM), *=0.05>p>0.01, **=p<0.01. L1 = mother tongue, L2 = second language learn at later stage. Map 2, which correspond to the largest part of the P100 time windows, extends between 100 and 150 ms after stimulus presentations. Map 4. Which corresponds to N1 extends between around 150 and 300 ms after stimulus presentations.